# The Role of Resilience and Basic Hope in the Adherence to Dietary Recommendations in the Polish Population during the COVID-19 Pandemic

**DOI:** 10.3390/nu13062108

**Published:** 2021-06-19

**Authors:** Beata Sińska, Mariusz Jaworski, Mariusz Panczyk, Iwona Traczyk, Alicja Kucharska

**Affiliations:** 1Department of Human Nutrition, Faculty of Health Sciences, Medical University of Warsaw, 27 Erazma Ciołka Street, 01-445 Warsaw, Poland; beata.sinska@wum.edu.pl (B.S.); iwona.traczyk@wum.edu.pl (I.T.); alicja.kucharska@wum.edu.pl (A.K.); 2Department of Education and Research in Health Sciences, Faculty of Health Sciences, Medical University of Warsaw, 61 Żwirki i Wigury Street, 02-091 Warsaw, Poland; mariusz.panczyk@wum.edu.pl

**Keywords:** COVID-19, dietary habits, psychological adaptation, observational study, personality, psychological resilience, life stress

## Abstract

(1) Background: The COVID-19 pandemic exerts a negative influence on dietary behaviors, which may lead to health deterioration. Dietary behaviors may be determined by psychological characteristics, such as basic hope and resilience, which facilitate the effective adjustment to new difficult conditions. The professional literature includes no research on the role of basic hope and resilience in the context of undertaken dietary behaviors in the situations of mental load associated with pandemics. The study aimed at the description of the dietary behaviors of individuals with various intensities of the discussed psychological characteristics (basic hope and resilience); (2) The observational cross-sectional online questionnaire study was conducted with the participation of 1082 adult Polish inhabitants. Three psychological scales were used: PSS-10, the Brief Resilient Coping Scale (BRCS) and BHI-12 questionnaire. The assessment of the adherence to dietary recommendations was performed with the present authors’ Dietary Guidelines Adherence Index (DGA Index); (3) Results: The value of DGA Index was variable depending on the psychological profile of study participants. The highest adherence to the principles of appropriate nutrition was observed in individuals characterized by the ability to cope with difficult situations and those who quickly adapted to new changing circumstances. The DGA Index values became poorer with the deterioration of the coping ability as regards stress and mental load; (4) Conclusions: Nutritional education during pandemics should encompass the psychological profile of the patients. It requires the implementation of a different psychodietetic approach which will facilitate a more effective introduction of a well-balanced diet.

## 1. Introduction

The state of epidemic threat was introduced in Poland in the middle of March 2020 because of the COVID-19 pandemic. The first case of SARS-CoV-2 infection was reported in Poland on the 4th of March. The state of epidemic threat was introduced between the 14th and 20th of March, and the state of epidemic emergency was declared on the 20th of March pursuant to a Regulation of the Minister of Health. All Polish universities closed down on the 12th of March. Public gatherings were banned, the movement of persons and access to public spaces were restricted. Tight restrictions which almost completely banned the use of public spaces remained in force until the 20th of April [1].

The COVID-19 pandemic has considerably changed the functioning of many people, not only in the professional, but also in the private aspect. The rapid transmission of SARS-CoV-2 (Severe acute respiratory syndrome coronavirus 2) [2] contributed to the introduction of numerous restrictions, e.g., social distancing, isolation, home confinement [3], remote work [4] and education [5]. Such changes disrupted the activities of daily living and family life of the majority of people. It contributed to the escalation of conflicts and a negative influence on the quality of relations in close relationships and the necessity to reorganize responsibilities at home and the rules of child care. In many cases combining remote work and family responsibilities was hindered or even impossible [6].

Changes in the activities of daily living associated with the COVID-19 pandemic were also observed with regard to dietary behaviors [7]. The results of research conducted to date indicated the increase in consumption during the pandemic. Sidor et al. reported that food consumption and was higher and snacking was more frequent in a considerable percentage of individuals which might cause body weight gain [8]. Similarly, Górnicka et al. reported that over one-third of the respondents declared an increase in total food consumption and the consumption of confectionery, while almost every fifth respondent—higher alcohol consumption. The authors also emphasized that quarantine might exert a bidirectional effect, positive and negative, on changes in dietary habits, because of an increased amount of time spent at home. A particularly unfavorable effect of the COVID-19 pandemic on the dietary habits was observed in adults over 40, persons living with children, persons living in regions with higher GDP and those who had not eaten at home prior to the pandemic [9].

Changes in dietary habits may lead to the deterioration of the health status of isolated individuals depending on the time of isolation, its degree and the health status of the isolated people [10]. Therefore, it is crucial to identify the factors which might influence the fact of undertaking appropriate dietary behaviors in difficult situations (e.g., during pandemics). It needs to be emphasized that the dietary behaviors of people during pandemics may depend on psychological factors (e.g., a high level of anxiety and stress associated with the pandemic) [11]. Personality traits may also play a key role, as they facilitate the effective adjustment to new circumstances. The analysis of nutritional behaviors during isolation is a relatively new issue. Therefore, it seems justified to develop a theoretical model to characterize the correlations between personality traits and nutritional behaviors. Resilience may be a crucial trait in this model [12]. It was defined as a trait which facilitated the adjustment to new conditions and situations, particularly those which were difficult for a person [13]. The COVID-19 pandemic may certainly be such a situation. Moreover, some authors considered such a trait as motivating and mobilizing in terms of initiating the effective management of a difficult situation at a cognitive and emotional level [14]. Therefore, it may be assumed that the nutritional choices of highly resilient individuals would be more analyzed and adherent to the principles of healthy nutrition as opposed to the nutritional decisions of low-resilience individuals which may be directly modified by external factors, e.g., those associated with a high level of stress. Highly resilient individuals are considered to be characterized by more balanced functioning in traumatic and difficult situations [14]. Furthermore, individuals with high resilience find it easier to cope with stress, because they perceive stressful situations as a challenge to face. Therefore, they are able to deal with negative emotions and experience positive ones [15,16] which facilitates the determination of an effective solution [17,18]. Preliminary research showed that resilience was positively correlated with well-being in people [19]. There is a paucity of studies which analyzed the role of resilience in the context of undertaken dietary behaviors in a stress-inducing environment, e.g., during a pandemic.

The discussion of personality resources which may be helpful in coping with mental overload during the COVID-19 pandemic should also tackle the issue of basic hope [20]. According to Erikson [20] the feature influenced behaviors and beliefs concerning the meaning of events which occurred in human life. The feature constitutes the measure of confidence and motivation to undertake effective actions. Notably, no research has been published on the issue of the role of basic hope in human behavior, including dietary behaviors, during the COVID-19 pandemic. However, basing on the psychosocial concept developed by E. Erikson [20] it may be assumed that the discussed trait might be of particular significance in the context of experiencing social isolation combined with stress and anxiety concerning the future.

The present research was conducted with regard to the lack of available studies on the role of basic hope and resilience in the context of undertaken dietary behaviors in the situations of stress and mental load associated with pandemics. The research aimed at developing the characteristics of the dietary behaviors of individuals with various intensities of the discussed psychological traits (basic hope and resilience).

## 2. Materials and Methods

### 2.1. Design and Participants

The observational cross-sectional online questionnaire study was conducted in April and May 2020 in the inhabitants of Poland. The inclusion criteria were: respondents aged 18 and older capable of expressing informed consent to participate in the study. Individuals younger than 18 and those who did not consent to participate were excluded from the study. Nonprobability sampling of study participants was used. The sample size for this study was 1082 adult inhabitants of Poland.

### 2.2. Data Collection

The questionnaire was distributed via the Google forms web survey platform. The link to the questionnaire was shared via social media such as Facebook, Instagram, WhatsApp and the personal contacts of study group participants (the snowball method). The method of questionnaire distribution was selected due to the limited possibility of contacting the respondents directly because of the restrictions introduced by the Minister of Health in relation to the COVID-19 pandemic. An online survey is the recommended approach which facilitates rapid access to the study group while maintaining safety measures in such a situation [21]. The present research lasted from the beginning of April till the end of May 2020. It was a period of sudden changes in lifestyles related to the introduction of hard lockdown (12 March–20 April 2020).

### 2.3. Instruments

Psychological and nutritional variables were measured during the study. The psychological variables were assessed with the use of the following instruments:

The PSS-10 scale developed by Cohen, Kamarck and Mermelstein [22]. The scale is used to measure the perceived stress level. The present authors used the Polish adaptation of this scale developed by Ogińska-Bulik and Juczyński [13]. The PSS-10 includes 10 questions concerning various subjective feelings associated with problems, personal events, behaviors and coping methods. The test is used to assess the intensity of stress associated with one’s life situation over the past month. The scale is characterized by good accuracy and reliability with the Cronbach’s alpha coefficient estimated at 0.86 [18].

The Brief Resilient Coping Scale (BRCS) developed by Sinclair and Wallston [23] which measures resilience as a process. The present authors used the Polish adaptation of this scale developed by Piórowska et al. [12]. The scale consists of only 4 items. The BRCS may be useful in the identification of people who need support to develop their resilience. The accuracy of the scale is satisfactory. However, it requires further analyses. This psychological trait refers to the quality and satisfaction with life. The reliability of the BRCS is satisfactory. The Cronbach’s alpha coefficient is 0.625 [12].

The Polish version of the BHI-12 questionnaire was developed by Trzebiński and Zięba [24] for the measurement of basic hope understood as the belief of an individual concerning the organization and meaning of the world and its agreeableness. Basic hope refers to Erikson’s [20] concept of psychosocial development and is defined as general, early formulated conviction that the surrounding reality makes sense and is favorable. It is also a factor determining the constructiveness of human reactions to changes and breakthrough events, especially in situations in which irreversible loss may be incurred. The BHI-12 questionnaire includes 12 statements and is characterized by good accuracy and reliability with the Cronbach’s alpha coefficient estimated at 0.70 [24].

The assessment of the adherence to dietary recommendations was performed with the present authors’ Dietary Guidelines Adherence Index (DGA Index) developed on the basis of current “Healthy nutrition recommendations” for the Polish population published by the National Institute of Public Health—the National Institute of Hygiene [25]. They were developed basing on the review of professional literature concerning the influence of individual dietary components and types of diet on human life and the review of the recommendations of scientific societies dealing with this issue worldwide. Particular attention was paid to domestic and international studies concerning the health status and analyzing the threatening factors, including the results of the Global Burden Diseases Study [26]. Basing on the data we determined the most important diet-related risk factors. Study results were also used to specify the order of factors which contributed to the most considerable reduction in healthy life years in the Polish population. Consistently with study results, diet-related factors were presented in the order starting from ones which were the most significant for maintaining health in the new recommendations. The recommendations were graphically presented as a plate filled with various products with the indication of the recommended portions of individual groups of products in the daily diet. Additionally, groups of products were divided into the following categories: “Eat less”, “Eat more” and “Replace” [25].

Basing on the above described recommendations and thorough literature analysis, a group of experts in dietetics indicated 18 groups of products and the recommended frequency of their consumption. Ten groups included products which should be consumed in higher amounts as they exerted a positive effect on the health (vegetables, fruits, whole grains, dairy products (no added sugar), legumes, fish, unsalted nuts, seeds, white meat, oils/margarines, water), and eight groups—products whose consumption should be limited or replaced with healthier alternatives (red and processed meat, sweets, salty snacks, sweetened drinks, refined grains, fast food, butter/ lard, processed cheese). The groups of products were used to develop a consumption frequency questionnaire. Basing on the questionnaire the respondents selected the frequency of the consumption of the individual groups of products during the pandemic. They could choose the following answers: “a few times a day”, “once a day”, “a few times a week”, “once a week”, “1–3 times a month” and “never”. One point was scored if the frequency of the consumption of a specific group of products adhered to the recommendations (Table 1). If a response revealed no adherence to the recommendations—0 points were scored. DGA Index value was expressed as the total score between 0 and 18 points. Higher DGA Index values were interpreted as a higher degree of adherence to dietary recommendations (0 points—a complete lack of adherence to the recommendations, 18 points—complete adherence to recommendations). The reliability assessment showed a satisfactory level of the internal consistency of the measurement (Cronbach’s alpha coefficient estimated at 0.67). The criterion, differential, or convergent validity of the DGA Index was not estimated.

The questionnaire also included questions regarding sociodemographic data, including: age, gender, place of residence, professional activity and the level of education. The Dietary Guidelines Adherence Index is available as Appendix A (See: Appendix A).

### 2.4. Ethical Considerations

Prior to the study the participants were informed that it was anonymous and the data were confidential. No personal data or computer IP were collected. Due to the anonymous character of the questionnaire and no possibility to follow sensitive data the study required no approval of the Bioethics Committee.

### 2.5. Data Analysis

Quantitative and categorical variables were described with the methods of descriptive statistics. The following measures were determined for quantitative variables: a central tendency (the means and 95% confidence interval, and the median), dispersion (standard deviation, interquartile range), location (upper and lower quartile). The number (N) and frequency (%) were determined for categorical variables.

During the first stage data clustering with k-means method was used to distinguish groups of participants varying in terms of psychological characteristics. The algorithm of 10-fold cross-validation was used, which allows the automatic determination of the number of data clusters. Selecting and distinguishing groups of similar objects in three clusters was performed during the analysis. Non-hierarchical clustering algorithm was implemented on the basis of the values calculated for three indices: stress, basic hope and resilience. Three groups of participants were distinguished: G1 (respondents with a moderate level of basic hope and resilience, but very stressed), G2 (respondents with a moderate level of basic hope and resilience) and G3 (respondents with a high level of basic hope and resilience). In case of groups 2 and 3, the level of stress was typical of the general population in a normal situation. In other words, those two groups were characterized by the level of stress similar to that before the pandemic.

During the second stage of the analysis, a comparison between the distinguished groups was performed depending on the dependent variable: the chi-squared test and one-way analysis of variance (one-way ANOVA) with the post hoc Tukey’s HSD (Honestly Significant Difference) test. The effect size of the observed difference was estimated with η^2^ assuming the following threshold values: 0.01 = small, 0.06 = medium, 0.13 = large effect size.

All calculations were performed with STATISTICA TM 13.3 software (TIBCO Software, Palo Alto, CA, United States). The *p*-level of < 0.05 was considered statistically significant in all analyses.

## 3. Results

### 3.1. Demographic Characteristics

A total of 1082 adult inhabitants of Poland participated in the study. The average age of study participants was 31.6 (SD = 11.98). Women constituted the marked majority of the respondents (*N* = 934, 86.3%). Selected demographic characteristics of the study group are presented in Table 2.

### 3.2. Psychological Characteristics

The level of stress measured with the PSS-10 appeared to be at a moderate level in the study group (M = 22.4, SD = 4.52, min-max: 6.0–40.0), while the distribution of the variable was distinctly symmetrical (skew: 0.01).

The level of resilience in the study group, measured with the BRCS, was slightly over the average for the population (M = 14.1, SD = 2.64, min-max: 4.0–20.0), which resulted in a slight left-sided asymmetry of distribution (skew: −0.48).

The level of basic hope measured with the BHI was also slightly over the average noted for the population reference ranges (M = 40.7, SD = 6.03, min-max: 20.0–56.0), which resulted in a slight left-sided asymmetry of distribution (skew: −0.31).

All scores obtained after the measurement with the three psychometric tools were converted to standard ten (sten scores) according to the reference ranges for the Polish population. It constituted the basis for the assignment to groups characterized by various levels of the studied psychological traits (Table 3).

### 3.3. Group Psychological Profile

Three groups of study participants distinguished with k-means clustering were characterized by the following profile of psychological traits (Table 1):

Respondents from G1 were characterized by a high intensity of stress, while G2 and G3 individuals were characterized by the levels of stress typical of the general population. However, the level of stress was typical of the conditions prior to the pandemic. In other words, the respondents from groups G2 and G3 did not assess the pandemic situation as a more or less stressful compared to the situation before the pandemic. Moreover, the respondents from G1 were characterized by the moderate levels of basic hope and resilience, i.e., levels typical of the general population. Therefore, this group reflected the attitude of a standard inhabitant experiencing a high level of stress associated with the pandemic. Conversely, G2 reflected the attitude of a standard inhabitant who experienced moderate stress levels which was unrelated to the pandemic. G3 was a specific group of individuals characterized by a high level of basic hope and resilience. It included the respondents who found it very easy to cope with difficult situations and quickly adapted to new changing conditions.

The groups were significantly different as regards three analyzed psychological traits (Table 4).

### 3.4. Intergroup Differences in Dietary Guidelines Adherence Index

The comparative analysis of three groups of study participants revealed that they were significantly different as regards the adherence to dietary recommendations (Figure 1). Post hoc analysis showed that the average value of DGA Index was significantly higher in G3 compared to G1 (12.6 vs. 11.8; HSD test: *p* < 0.001) and G2 (12.6 vs. 12.1; HSD test: *p* = 0.010). However, no significant differences occurred in the average DGA Index between G1 and G2 (11.8 vs. 12.1; HSD test: *p* = 0.348).

Some significant intergroup differences were noted in the assessment of the adherence to dietary recommendations in terms of the frequency of the consumption of individual product groups. The detailed results of comparative analysis were collected in Table 5.

## 4. Discussion

The study was conducted to determine the quality of the diet consumed during the pandemic with the assessment of the degree of adherence to the principles of appropriate nutrition (DGA Index). Similarly to other countries, it was recommended to stay at home and avoid social contact as the basic rule of limiting the exposure to the spreading virus during the COVID-19 pandemic in Poland [27]. The limitations resulted both in positive and negative changes in lifestyle and dietary habits [9]. A lot of evidence was collected to demonstrate that healthy lifestyle, including the consumption of appropriate food, played a key role in building resistance to diseases and maintaining good health status. Normal sleep, moderate physical activity, avoiding stress and consuming food rich in nutrients may naturally support the immune system, which is of particular importance in case of viral diseases, such as COVID-19 [28].

The present study showed that the value of DGA Index varied depending on the psychological profile of study participants. The highest adherence to the principles of appropriate nutrition was observed in case of G3 individuals who were characterized by the ability to cope with difficult situations and quick adaptation to new changing circumstances. The results confirmed the key role of psychological variables in making decisions associated with dietary habits in a difficult situation, such as a pandemic. Particular attention was paid to resilience which determines the ways of coping with difficult situations in life via effective dealing with stress and negative emotions [12,13,14]. It was noted that individuals with a high intensity of this psychological trait achieved higher values of DGA Index, i.e., a better degree of adherence to healthy diet. The observations are consistent with the results of psychological research which revealed that high levels of resilience promoted undertaking behaviors that were favorable for one’s health [13,29], which is of particular importance in the pandemic context. The obtained results were also the first ones to reveal the significance of resilience with reference to dietary behaviors during the pandemic. Notably, high levels of resilience may protect people from “the trap of eating”, reduce the tendency towards emotional eating and facilitate the mobilization towards undertaking action in difficult situations [29,30].

Basic hope was another analyzed psychological variable. It is one of the most important motivators of human activity which allows the interpretation of current events, especially if a person attempts at predicting their direction and consequences [18,20,24]. Considering the characteristics of this psychological variable it was assumed that it would support undertaking health-promoting dietary behaviors during the pandemic. This theoretical assumption was confirmed in the present study which demonstrated that basic hope combined with high resilience contributed to undertaking dietary behaviors adherent to recommendations. No research was published on the analysis of the role of basic hope in the context of dietary behaviors which suggests the necessity to conduct further empirical studies in this area.

The present study indicated the need to analyze the mutual system of psychological characteristics, and not individual traits, with reference to dietary behaviors, because psychological characteristics may appear stronger or weaker when combined. It was demonstrated that stress was not a factor which determined dietary behaviors. American research revealed that stress, especially at the initial stages of the pandemic, exerted no significant influence on changes in dietary habits [31] which is in line with our research. However, some authors emphasized the importance of mental stress on dietary behaviors during the pandemic [32]. Notably, stress may cause an individual response to the change in dietary behaviors. The above mentioned change may be short-term or long-term. It depends on the psychological mechanisms of an individual. Some people may tend to consume higher quantities of food is a stressful situation (about 40% of the general population), while others considerably limit the amount of food consumed (about 40% of the population). It is believed that in case of 20% of the population stress does not contribute to changes in dietary habits [33]. The underlying cause has not been fully elucidated. It is assumed that the emotional status and the psychological profile is of importance here. The level of experienced stress may be modified by the mutual system of traits, such as basic hope and resilience which are psychological characteristics determining the effectiveness of coping with difficult situations. It may be significant in the context of the personalization of dietary counseling. Moreover, personalized counseling should also comprise the aspect of emotional eating which has a negative impact on dietary behaviors during the COVID-19 pandemic [34].

The study indicated that dietary counseling during difficult and stressful times, e.g., during a pandemic, should encompass the current psychological status of patients/clients. It is particularly important to determine the degree to which the individual copes with the current situation. It was demonstrated that persons with high resilience and basic hope were characterized by the dietary behaviors which were the closest to the recommended ones. However, they did not fully adhere to those recommendations. Therefore, the dietary counseling of such individuals should be based on the identification of the discussed psychological characteristics and strengthening them. Diet interviews should encompass questions concerning the level of perceived stress and behaviors related to emotional eating. It is also important to assess the self-efficacy of clients/patients with regard to the preparation of meals comprising the principles of healthy eating. As regards such a group of individuals, the dietician should use tools aiming at strengthening the sense of accomplishment, emphasize behaviors which are in line with current dietary recommendations and provide support in the modification of behaviors which should be changed. It is a context in which the dietician is a tutor showing the way and the dietician-patient relation is based on partnership.

In case of patients with low or moderate levels of resilience, nutrition education should be focused on to the analysis of methods used by the patient to cope with stress, especially those which are directly related to eating habits. The dietician should also discuss the role of nutrients in strengthening the functioning of the nervous and immune systems and alleviating stress related to the experience of negative events. If a patient cannot cope with stress, the dietician should suggest a psychological consultation and the inclusion of psychological counseling into the therapy, particularly if the problem exceeds the standard competences of the dietician. Notably, strong stress and negative emotions may constitute a significant barrier hindering diet modification in a certain direction. First, the patient should deal with negative experiences effectively in order to be able to undertake actions. Therefore, the dietician should be highly empathic, calm and reactive to every doubt expressed by the patient. It is important to adjust the quantity and quality of information adequately to the emotional status and cognitive capacity of the patient. It is worth remembering that cognitive capacity becomes more modifiable with stronger negative emotions [35]. In other words, a patient who experiences negative emotions will remember much less than one experiencing positive emotions. Moreover, suggested changes may seem more difficult to achieve and one’s own capacities may be perceived as poorer than in individuals with positive emotional status. Therefore, the dietician should adjust diet modification using the tools of motivational interviewing which will not only facilitate the reduction of the negative mental status of the patient, but also strengthen the self-evaluation in the patient [36].

The analysis of the adherence to dietary recommendations should focus on the consumption of individual groups of products, especially those which exert a positive effect on health. International guidelines concerning public health emphasize the significance of diet during epidemics. The diet should be based on fruits, vegetables, whole-grain products, low-fat milk products and healthy fats (olive oil and fish oils). The consumption of sweet drinks and processed food with high energy and salt content should be limited. [37].

The present study revealed that the frequency of consuming fruits and vegetables consistent with the recommendations was reported in slightly over 50% of the respondents. The percentage of G3 individuals characterized by the appropriate frequency of the consumption of those products was significantly higher compared to the remaining groups. However, it was still lower than the satisfactory level and was only 65%. Fruits and vegetables should be consumed daily in large amounts (min. 400–500 g daily) both fresh and frozen [38]. The favorable effect of fruits and vegetables was noted, for example they reduce the risk of some chronic diseases, such as cardiovascular diseases [39]. The consumption of fruits and vegetables was also investigated in terms of the potential benefits related to respiratory disorders [40] and inflammatory diseases [41].

The suitable consumption of whole-grain cereal products is an important element of health-promoting diet. The present results indicated that a considerable percentage of the respondents did not consume them at the recommended frequency. The percentage of individuals with the appropriate consumption of whole grains did not exceed 40% in G1 and G2, while in G3 it was significantly higher (47.5%), but still insufficient. The positive influence of whole grains on improving immune function is observed [42].

Currently available research indicated that vitamin D deficiency was associated with an increased risk of developing COVID-19 [43]. It is even more important to provide its suitable amounts with the diet during the pandemic. It may be achieved by the consumption of larger proportions of its sources—milk products and fish. Our study showed that the recommended level of dairy product consumption was reached by less than half of G1 and G2 respondents (49.2% vs. 45.7%, respectively). The percentage of individuals adhering to the recommendations in group G3 was significantly higher (55.6%), but, still, it could not be assessed as satisfactory. Fish consumption was even lower. The frequency compliant with the recommendations was noted in only one-third of G1 and G2 individuals (31.5 vs. 34.4), and in less than half of G3 individuals. The suitable frequency of fish consumption is particularly important in the context of viral diseases. It was demonstrated that fish oil increased antiviral response and inhibited virus replication [44]. It is possible that the increased dietary omega-3 consumption or its supplementation may promote the prophylaxis and treatment of COVID-19 via the reduction of the intensity of the inflammation (COVID-19) [45]. Therefore, omega-3 acids may exert a positive effect on inflammatory diseases [46].

Nuts and seeds are the element of healthy dietary pattern. They are characterized by a high content of unsaturated fatty acids, plant protein, a considerable amount of polyphenols, phytosterols, dietary fiber, and vitamins and minerals (folic acid, vitamin E, selenium, magnesium) [47], which may have a synergistic action and promote health, also by supporting the immune function. To date, no research has been conducted to assess the possible effect of nut consumption in combating viral or bacterial diseases or the potential complications of such infections [48].

Regardless of the psychological profile, over 90% of the respondents consumed water at the recommended frequency. Nevertheless, similarly to the remaining recommendations, G3 individuals were characterized by the highest adherence. Proper hydration is essential for healthy life. Water is necessary for maintaining cellular homeostasis. It transports nutrients to cells and removes the waste products of metabolism. Water facilitates the functioning of all transport systems enabling exchange between the cells, interstitial fluid and capillaries. Water supports the maintenance of vascular volume and enables blood circulation, which is essential for the functioning of all body organs and tissues. Therefore, appropriate hydration is necessary to provide the normal activity of the cardiovascular and respiratory systems, the digestive tract, reproductive system, kidneys, liver, brain and the peripheral nervous system [49]. Drinking water may strengthen the adaptive immune response by removing toxins from the body through the kidneys and sweat glands [50].

As regards unhealthy products, the adherence to recommendations was diversified between groups in case of sweetened drinks. A significantly lower percentage of persons consumed such drinks at a recommended frequency in group G1. The consumption of non-alcoholic beverages containing sugar is considered as the main factor contributing to the epidemic of obesity [51]. Moreover, non-alcoholic beverages reduce the feeling of satiety and the level of perceiving the sweet taste, leading to excessive energy consumption and. As a consequence, body weight increases and the risk of developing type 2 diabetes mellitus is higher [52,53].

It is worth noting that undertaking unhealthy eating behaviors (e.g., an increased consumption of high-energy snacks) during the pandemic was associated with the experience of strong negative emotions (e.g., anxiety) and restrictions (e.g., lockdown) [54]. Similar observations were reported by other researchers [32].

The present research was conducted to analyze the role of basic hope and resilience in the context of undertaken dietary behaviors in the situations of mental load associated with the pandemic. It is a new approach to the issue which has not been discussed in the literature yet.

However, there are some limitations. Firstly, the study did not comprise the changes of dietary behaviors over time. The measurements were performed at one time point. Considering the above, the obtained psychological characteristics should not be discussed in cause-and-effect categories. Moreover, the direction of the changes of dietary behaviors is unknown. Therefore, the determination of the principles of dietary counseling was limited to outlining the methods of cooperation based on the “here and now” approach. The authors also did not aim at the discussion of the psychological traits in the context of causes. The present authors’ Dietary Guidelines Adherence Index (DGA Index) developed on the basis of current “Healthy nutrition recommendations” for the Polish population published by the National Institute of Public Health—the National Institute of Hygiene is another important limitation of the study [24]. The index is based on the declarations of the respondents, so it should be interpreted with caution. The researchers did not verify whether the declarations of the respondents regarding their dietary behaviors were true. The principle of confidence was implemented in this study. Potential confounding factors (e.g., smoking status) constitute one of the important limitations of the present study. The study did not involve the analysis of stimulant use by the respondents and their influence on dietary behaviors during the pandemic. Further research should comprise such factors. Moreover, the study was conducted via the Internet, so only respondents with technology access could be included. Women from big cities were the dominant group in the study. Therefore, the interpretation of obtained data in terms of the general population should be rather cautious. Further research should be conducted with a larger percentage of men to verify the obtained correlations.

## 5. Conclusions

Nutrition education during pandemics should comprise the psychological profile of the patient. It requires the use of a different psychodietetic approach which may facilitate a more effective introduction of a well-balanced diet. Appropriate dietary habits are more important during pandemics than at any other point of time, due to their role in strengthening the immune system. It is now a priority to have a good life and live in a healthy way. A well-balanced diet concentrated on fruits, vegetables, whole-grain products, plant and animal protein and healthy fats is the best way to obtain all nutrients necessary for good health and appropriate functioning of the immune system.

## Figures and Tables

**Figure 1 nutrients-13-02108-f001:**
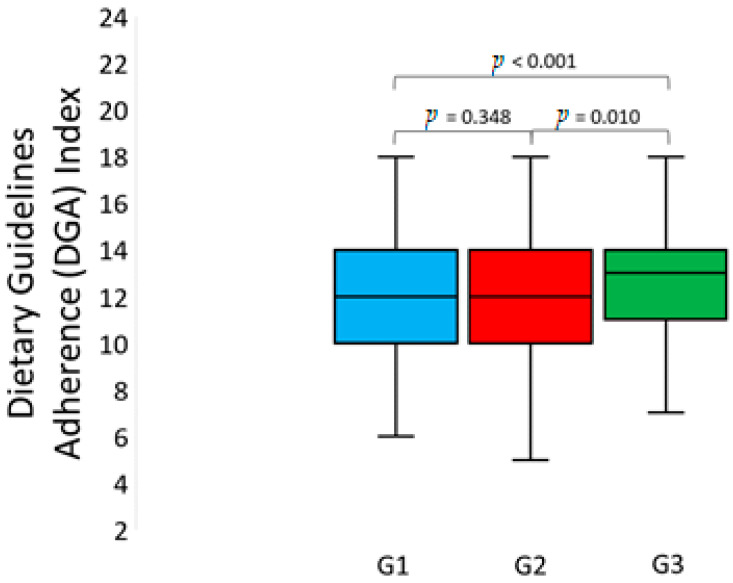
Dietary Guidelines Adherence (DGA) Index in groups with different psychological profiles (one-way ANOVA: F_(2, 1079)_ = 10.104, *p* < 0.001, eta^2^ = 0.018).

**Table 1 nutrients-13-02108-t001:** The components of the Dietary Guidelines Adherence Index.

Group of Products	Recommended Frequency of Consumption
Groups of healthy products	
Vegetables	a few times a day
Fruits	once a day or more
Whole grains	once a day or more
Dairy products (no added sugar)	once a day or more
Legumes	a few times a week or more
Fish	once a week or more
Unsalted nuts, seeds	a few times a week or more
White meat	a few times a week or less
Oils/margarines	a few times a week or more
Water	once a day or more
Groups of unhealthy products	
Red and processed meat	once a week or less
Sweets	once a week or less
Salty snacks	once a week or less
Sweetened drinks	once a week or less
Refined grains	once a day or less
Fast food	once a week or less
Butter/lard	once a week or less
Processed cheese	once a week or less

**Table 2 nutrients-13-02108-t002:** Characteristics of the study group (*N* = 1082).

Age (years)	The value of variable
M ± SD (95% CI)	31.6 ± 11.98 (30.9–32.3)
Mdn ± IQR/2	27.0 ± 9.00
Min-max	18.0—82.0
Gender, *N* (%)	The value of variable
Male	148 (13.7)
Female	934 (86.3)
Place of residence, *N* (%)	The value of variable
Village	181 (16.7)
Small town	193 (17.8)
Town	178 (16.5)
Big city	530 (49.0)
Education, *N* (%)	The value of variable
Primary/vocational	13 (1.2)
Secondary	355 (32.8)
Tertiary	714 (66.0)

M—mean, SD—standard deviation, CI—confidence interval, Mdn—median, IQR—interquartile range.

**Table 3 nutrients-13-02108-t003:** The levels of the analysed psychological characteristics according to the norms for the Polish population.

Psychological Characteristic	Level	*N*	%
Stress			
	low	381	35.2
	moderate	348	32.2
	high	353	32.6
Resilience			
	low	270	25.0
	moderate	483	44.6
	high	329	30.4
Basic hope			
	low	301	27.8
	moderate	439	40.6
	high	342	31.6

**Table 4 nutrients-13-02108-t004:** Analysis of intergroup differences in the mean levels of selected psychological characteristics.

	G1 (*N* = 352)		G2 (*N* = 334)		G3 (*N* = 396)		F_(2, 1079)_	*p*-Value *
	M	SD	M	SD	M	SD		
Stress	7.3	1.14	3.8	1.11	5.3	1.71	577.311	<0.001
Resilience	4.5	1.73	4.7	1.61	7.0	1.29	302.021	<0.001
Basic hope	4.3	1.59	5.0	1.58	7.2	1.37	373.917	<0.001

M—mean, SD—standard deviation. * one-way ANOVA.

**Table 5 nutrients-13-02108-t005:** Intergroup comparison of the frequency of consuming individual product groups with different psychological profiles.

Product Group	G1 (*N* = 352)		G2 (*N* = 334)		G3 (*N* = 396)		χ^2^_df = 2_	*p*-Value ^1^
	N	%	N	%	N	%		
Groups of healthy products								
Vegetables	185	52.6	192	57.5	259	65.4	13.029	0.001
Fruits	212	60.2	201	60.2	272	68.7	7.778	0.020
Whole grains	135	38.4	130	38.9	188	47.5	8.093	0.017
Dairy products (no added sugar)	173	49.2	152	45.7	220	55.6	7.442	0.024
Legumes	109	31.1	105	31.4	131	33.1	0.403	0.818
Fish	111	31.5	115	34.4	180	45.6	17.650	0.000
Unsalted nuts, seeds	155	44.0	160	47.9	223	56.5	12.160	0.002
White meat	331	94.0	316	94.6	372	93.9	0.168	0.919
Oils/margarines	288	81.8	279	83.5	328	82.8	0.358	0.836
Water	329	93.5	304	91.0	381	96.2	8.355	0.015
Groups of unhealthy products								
Red and processed meat	294	83.5	272	81.4	309	78.0	3.735	0.155
Sweets	125	35.5	133	39.8	160	40.4	2.169	0.338
Salty snacks	290	82.4	289	86.5	342	86.6	3.275	0.195
Sweetened drinks	242	69.0	262	78.7	309	78.0	11.243	0.004
Refined grains	351	99.7	334	100.0	396	100.0	2.076	0.354
Fast food	329	93.5	321	96.1	379	96.0	3.397	0.183
Butter/lard	349	99.2	331	99.1	390	98.5	0.942	0.624
Processed cheese	158	44.9	146	43.8	168	42.4	0.466	0.792

^1^ chi-squared test.

## Data Availability

The data presented in this study are available on request from the corresponding author. The data are not publicly available due to psychological data.

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
