# Peer review of "The Role of Resilience and Basic Hope in the Adherence to Dietary Recommendations in the Polish Population during the COVID-19 Pandemic"

_nutrients, 2021, doi:10.3390/nu13062108_

Round 1

Reviewer 1 Report

The manuscript submitted for consideration at Nutrients by Sinska et al., titled: “The role of resilience and basic hope in the adherence to dietary recommendations in the Polish population during the Covid-19 pandemic”, is aiming at investigating how resilience and positivism as expressed by basic hope can affect adherence to dietary recommendations in Polish individuals during the COVID-19 pandemic. This is an interesting topic and most pertinent to the current times that could provide useful insights as per strategies to improve adherence to dietary recommendations during a pandemic especially given the fact that restrictive measures applied during pandemics can have a negative impact on proper nutrition and subsequently long- and short-term weight gain and overall disease risk. The manuscript is well organized and written, and flows logically. The
reviewer would like to make the following points to the be considered by the authors for manuscript improvement.

1. What was the sample size that ended-up being used the authors mention 1082 in the results but consider specifying it also in the 2.1 section of Materials and Methods.
2. Did the authors consider potential confounding factors such as smoking status?
3. Was the questionnaire used validated?
4. Are parameters such as sleep, marital status, companionship, social networks, technology access, financial condition, and access to foods considered?
5. What was the level of access to unhealthy/comfort food the participants had? Was this parameter considered in the analysis?
6. It is helpful that the authors provide geographical and socioeconomic status, education status in addition to age/gender distribution information of the participants. However, there are some unbalanced distributions such as high female vs. male representation and high urban vs other locations in the participants. How have the authors considered biases of results due to these imbalances?
7. Table 3. First column second row: should read “stress” as opposed to “tress”?
8. Figure 1: Please provide key to color coding in the caption.
9. Consider enriching the literature on the including the dimension of comfort food and the psychosocial aspect of food and stress in the discussion.

Author Response

Dear Reviewer of  Nutrients

Thank you very much for all your comments and the opportunity to improve the manuscript. We have been working to introduce the comments provided by Reviewers. We hope that the present version of the manuscript will meet the expectations of the Reviewers of Nutrients. The changes within the revised manuscript were highlighted (underlined and in yellow). Once again, thank you for your valuable comments and attention given to our paper.

Please see our answer in the attached file.

Best regards,
Mariusz Jaworski

Reviewer 2 Report

This is an interesting article considering current global situation. The scientific rationale and the conceptual design are of relevance, and the article was conceived correctly. Introduction section has been adequately presented.

However, there are some comments to be address:

  • Introduction section has been adequately presented.
  • Please report the COVID-19 epidemiology trends during the period of analysis;
  1. a) The full version of the questionnaire should be available as Appendix.
  2. b) The authors should justify the start and end date of the web-survey.
  3. Did the authors check the health status of the participants? Did some of them were diagnosed with COVID19, or if one of the family members (or surrounding environment e.g., neighbors) was diagnosed?
  • A) figure n 1 is redundant and not very significant, please remove. B) The table n. 5 should be transformed into a figure.
  • Discussion:
  1. the discussion is too long. Authors should remodel it, removing the parts that are not useful to support or refute the results;
  2. the phrase "Multiorgan failure is one of the main causes of death in patients with the severe 409 course of COVID-19. It results from the exaggerated reaction of the immune system to the 410 contact with a pathogen inducing the cytokine storm." is redundant and not contextualized;
  3. the phrase "The recommendation concerning water was the best adhered to among all recom-425 mendations concerning healthy nutrition." it is not contextualized; it is not supported by the data and there are no recommendations on adequate consumption according to guidelines.
  • Authors should check the typos also present in the table. Please read carefully and correct them.
  • English language should be revised.

Author Response

Dear Editor of  Nutrients

Thank you very much for all your comments and the opportunity to improve the manuscript. We have been working to introduce the comments provided by Reviewers. We hope that the present version of the manuscript will meet the expectations of the Reviewers of Nutrients. The changes within the revised manuscript were highlighted (underlined and in green). Once again, thank you for your valuable comments and attention given to our paper.

Please see our answer in the attached file

Best regards,
Mariusz Jaworski

Round 2

Reviewer 1 Report

There reviewer believes that the authors addressed comments in a satisfactory manner.